# Influence of upper limb training and analyzed muscles on estimate of physical activity during cereal grinding using saddle quern and rotary quern

Michal Struška[1]*, Martin Hora[1], Thomas R. Rocek[2], Vladimír Sládek[1]

1 Faculty of Science, Department of Anthropology and Human Genetics, Charles University, Prague, Czech Republic, 2 Department of Anthropology, University of Delaware, Newark, Delaware, United States of America

* struska1408@gmail.com

**Data Availability Statement:** All files are available from the Figshare database (DOI: 10.6084/m9.figshare.13265468).

## Abstract

Experimental grinding has been used to study the relationship between human humeral robusticity and cereal grinding in the early Holocene. However, such replication studies raise two questions regarding the robusticity of the results: whether female nonathletes used in previous research are sufficiently comparable to early agricultural females, and whether previous analysis of muscle activation of only four upper limb muscles is sufficient to capture the stress of cereal grinding on upper limb bones. We test the influence of both of these factors. Electromyographic activity of eight upper limb muscles was recorded during cereal grinding in an athletic sample of 10 female rowers and in 25 female nonathletes and analyzed using both an eight- and four-muscle model. Athletes had lower activation than nonathletes in the majority of measured muscles, but except for posterior deltoid these differences were non-significant. Furthermore, both athletes and nonathletes had lower muscle activation during saddle quern grinding than rotary quern grinding suggesting that the nonathletes can be used to model early agricultural females during saddle and rotary quern grinding. Similarly, in both eight- and four-muscle models, upper limb loading was lower during saddle quern grinding than during rotary quern grinding, suggesting that the upper limb muscles may be reduced to the previously used four-muscle model for evaluation of the upper limb loading during cereal grinding. Another implication of our measurements is to question the assumption that skeletal indicators of high involvement of the biceps brachii muscle can be interpreted as specifically indicative of saddle quern grinding.

## Introduction

The experimental reconstruction of habitual tasks of past populations is often used to test hypotheses linking skeletal markers of activity with the behavior of past humans [1–4]. Aside from animal models, it is impractical to observe developmental changes in human skeletal morphology as a result of activity patterns; thus, measurements of muscle activity in modern

**Funding:** This study was supported by the Czech Science Foundation (GACR, www.gacr.cz, project number: 18-16287S). The funding was awarded to M.H. One of the authors (M.H.) has been also supported by Charles University Research Centre program No. 204069. The funders had no role in study design, data collection and analysis, decision to publish, or preparation of the manuscript.

**Competing interests:** The authors have declared that no competing interests exist.

subjects replicating past activities serve as a route for inferring the impact of prehistoric behaviors. Two significant issues underlie such experimental replicative experiments: the choice of appropriate modern experimental subjects, and the choice of which muscle activity to measure.

The first issue raises the question of the applicability of experiments using modern sedentary and physically inactive populations whose lifelong pattern of physical exertion differs drastically from the heavy physical workload of the prehistoric populations of interest. These modern groups are typically assumed to be inappropriate for such experimental comparison with prehistoric samples, and instead studies of modern athletes are often considered a starting point for such research (e.g. [5–7]). However, while modern athletes probably do more closely resemble the physical condition of most prehistoric groups, here we consider whether comparisons among the physical demands of various prehistoric physical activities may not require restriction of the modern test sample to trained athletes if the *relative* differences between the activities are reflected in *relative* differences between experimental groups, even non-athletic ones.

A similar question relates to the choice of muscular data needed to conduct comparative studies of alternate activity patterns. Ideally, assessments of musculoskeletal consequences of physical exertion should measure as many of the muscles that are potentially recruited during the task as possible. However, experimental reconstruction using electromyography (EMG) often analyzes a limited set of muscles [3,4]. The number of muscles analyzed in experimental reconstructions might be limited by the number of available sensors and the duration of experimental sessions. Reducing the set of analyzed muscles might be inevitable, but the effect of reducing the investigated upper limb muscles to a small set has not been explored. Thus, it is useful to examine if broad sampling of muscle activity is necessary, or whether smaller selective samples of relevant major muscles can provide equivalent information. Finally, when consistent results are found, we can compare these against skeletal inferences of past activity patterns.

In this study we examine these two questions in the context of research on muscle activation associated with alternative patterns of cereal grinding, comparing reciprocal grinding using a saddle quern to rotary grinding with a rotary quern. Cereal grinding has previously been experimentally reconstructed to estimate the effectiveness of early agricultural grinding tools [4], the influence of different grains on the resulting product [8], and upper limb loading during grinding [4]. The pattern of upper limb loading during grinding was used to explain changes in humeral robusticity asymmetry among humans in the early Holocene [4,9]. This research was based on experimental grinding activities by a sample of modern female nonathletes.

Two recent papers on experimental approaches to prehistoric activity patterns have strengthened the argument for the use of modern athletes specialized in sports involving motions similar to the reconstructed archaeological tasks [5,6]. The influence of a modern athletic sample on experimental results was shown in [6], where athletes were able to throw replicas of Middle Paleolithic spears farther than previously believed based on a wider range of subjects. On the other hand, we don't know what influence an athletic sample would have on the previous experimental reconstruction of cereal grinding [4]. Even less clear is the influence of different levels of athletic training on the *relative* patterns of muscle use or effectiveness during these alternative grinding activities.

During physical activity, such as cereal grinding, muscle force stresses bone, to which the bone responds by increasing its resistance against the stress [10,11]. The magnitude and location of muscle force produced during cereal grinding might suggest where bone adaptation would happen. Muscle force is often approximated using the magnitude of muscle activation

[3,4], which describes the neural input which causes muscle contraction [12]. Muscle activation is measured using EMG. The relationship of muscle activation and muscle force is further influenced by cross-sectional area of the muscle [13,14], the length and the pennation angle of contracting muscle fibers [15], and the velocity of contraction [16].

Previous analysis of muscle activation during cereal grinding used female nonathletes [4]. However, early Holocene females had extensive experience with grinding, which western industrialized females lack. Experimental and ethnographic evidence suggest that early Holocene females may have spent 1.5–7 h/day grinding on saddle quern or 1.3 h/day grinding on rotary quern [4,8,17,18]. Daily repetition of grinding movement in early Holocene females could have had effects on muscle activation, as observed in contemporary manual workers [19–21]. Early agricultural females involved in daily repetition of cereal grinding might have develop more efficient grinding movement which would require less muscle force or which would require activity of different muscles than are used by modern females that are less involved in upper limb tasks. Therefore, skeletal adjustments in early agricultural females could occur in different places or to lower extent than inferred from contemporary female nonathletes. To account for experience, grinding could be studied in populations that still use manual grinding tools [17,18,22–24]. To account for differences between subjects would mean to have both grinding tasks done by the same subjects, who would be skilled in both grinding methods. Contrary to this ideal situation, the populations using manual grinding tools do not use saddle and rotary querns concurrently. Therefore, females from contemporary populations would not have experience with grinding on both saddle and rotary querns. As an alternative, the subsistence behavior of past populations could be studied in athletes, as was recently reviewed for reconstructions of other past human activities [5].

Cereal grinding does not have direct equivalent in any athletic discipline, but there are characteristics in which habitual grinding of early Holocene females resembles rowing. Contemporary female rowers have level of humeral robusticity similar to early agricultural females [25]. In both grinding and rowing, the upper limbs are involved in repetitive movement pattern. Duration of a rowing cycle can be as low as 1.6 s [26], which is similar to the 1.4 s cycle used for rotary quern grinding in a previous experimental study [4]. Professional rowers spend 11.2 h/week rowing [27], which is close to the estimated time for rotary quern grinding (9.1 h/week) [4]. These similarities between rowers and grinding females suggest that rowers might help us understand the effect of cereal grinding on human muscle activity better than the previously used nonathletes.

In addition to the question of the appropriate modern population sample for experimental testing is the issue of the choice of muscles to measure. The relationship between cereal grinding and muscle activation during grinding was previously measured in four muscles with insertion on the upper limb (anterior deltoid, infraspinatus, pectoralis major, and long head of the triceps brachii) [4]. While the previous study selected four prime movers of the upper limbs, more muscles may be necessary to see the effect of muscle forces that control shoulder and elbow motions during cereal grinding. The muscles examined in the previous study are mainly responsible for shoulder flexion and rotation and elbow extension [4]. By adding the muscles with other functions, such as shoulder extension and elbow flexion, we may estimate upper limb loading during a greater portion of shoulder movements.

Furthermore, the analysis of the activation of a larger set of upper limb muscles would indicate in which muscle insertions we may expect entheseal changes associated with cereal grinding on saddle and rotary quern. This could help connect entheseal changes with changes of grinding technology during the adoption and intensification of agriculture. In addition, we could test the activation of the muscles that were suggested to be active during cereal grinding based on entheseal changes, such as the deltoid or biceps brachii during saddle quern grinding

[28]. Thus, measures of muscle activation may assist in osteological interpretations of prehistoric activities.

As the first goal, we aim to compare muscle activation during grinding in athletes and nonathletes using electromyography. We expect athletes to have lower muscle activation during grinding than nonathletes. The second goal is to compare upper limb loading during saddle and rotary quern grinding estimated using the eight- and four-muscle models. We expect the eight-muscle model to show a lower difference between saddle and rotary quern grinding than the four-muscle model, as a result of the addition of the muscles presumed to be active during saddle quern grinding (biceps brachii, parts of the deltoid). Finally, we aim to suggest possible skeletal markers indicating saddle quern or rotary quern grinding in osteological remains.

## Materials and methods

### Sample

Two samples were used in this study. The first sample was an athletic sample consisting of 10 female rowers (age, 16.3 ± 0.8 years; body height, 170.0 ± 4.8 cm; body mass, 66.4 ± 8.4 kg). According to responses to a questionnaire, our athletic subjects spent on average 10.7 h/week rowing and 2 h/week doing other strenuous tasks involving the upper limb. The second sample consisted of 25 female nonathletes (age, 24.1 ± 2.8 years; body height, 166.3 ± 4.1 cm; body mass, 60.6 ± 9.2 kg) who were recruited mainly from the students of Charles University (Prague, Czech Republic). According to the questionnaire, nonathletes spent on average 3.4 h/week participating in sports and 1.6 h/week doing other strenuous tasks involving the upper limbs. Only right-handed individuals were included in the analyses. Right-handedness was estimated using the method of Oldfield [29]. Subjects participating in the study gave written consent. The study was approved by the Institutional Review Board of Charles University (Prague, Czech Republic), Faculty of Science (Approval Numbers: 2013/10 and 2017/28).

### Assessment of muscle activation

Eight muscles with insertion on upper limb were selected for the analysis: biceps brachii, anterior deltoid, middle deltoid, posterior deltoid, infraspinatus, pectoralis major, and lateral and long heads of the triceps brachii. The muscles were selected based on their role in movements that are used during saddle and rotary quern grinding. Saddle quern grinding consists mainly of shoulder and elbow flexion and extension. Rotary quern grinding includes shoulder and elbow flexion and extension as well as shoulder rotation, abduction, and horizontal adduction. Functions of the selected muscles are described in S2 Table.

Muscle activation was assessed using eight surface EMG sensors (Trigno Standard Sensor, Delsys, Natick, MA, USA), which were simultaneously put on the right biceps brachii; anterior, middle, and posterior deltoids; infraspinatus; pectoralis major; and lateral and long heads of the triceps brachii. The placement of the sensors is shown in S1 Fig. The skin was cleaned with isopropanol-soaked cosmetic pads [29]. Afterward, sensors were attached to the skin using a double-sided tape (Trigno Sensor Adhesive, Delsys, Inc.). The position and orientation of sensors were according to the recommendation of the Surface Electromyography for the Non-Invasive Assessment of Muscles (SENIAM) project [29]. An exception was the pectoralis major, for which the position of sensor was adopted from Ebben et al. [30], and infraspinatus, for which the position of sensor was adopted from Morris et al. [31]. The EMG signal was acquired with a frequency of 1,926 Hz and filtered using the Butterworth band-pass filter (20–450 Hz) implemented in Delsys Trigno sensors (Delsys, Natick, MA, USA). The raw EMG signal was full wave rectified and smoothed using the root mean square function (window length, 0.125 s; overlap, 0.0625 s) in EMGworks Software (Version 3.21, Delsys, Natick, MA, USA).

The average EMG curve was computed for each subject from at least three subsequent grinding cycles using the Cyclical Analysis script in EMGworks Software. The length of the average cycle was recalculated to 1,000 points using the Resample script in EMGworks Software. The EMG curve values were adjusted to maximum values obtained during maximum voluntary contraction (MVC) tests.

We obtained EMG activity at maximum voluntary isometric contractions using separated tests for each muscle following Konrad [32]. Maximum voluntary contractions tests are described in S2 Table. Each maximum voluntary contraction took approximately three seconds and was repeated three times. There was a pause of at least 30 seconds between each repetition. The MVC value was obtained using the highest value of the rectified and smoothed signal of all the measured contractions of given muscle. In every session, MVC tests were performed before grinding tasks. After MVC tests were finished, subjects were first shown how to use rotary quern and subsequently given time to try rotary quern grinding. This procedure took approximately five minutes. Subjects were introduced to saddle quern grinding after finishing the first rotary quern grinding task.

## Grinding

The experimental cereal grinding was performed using the Neolithic saddle quern (Fig 1A) and replica of the rotary quern (Fig 1B), which were previously used by Sládek et al. [4]. The saddle quern consists of a lower stationary and mobile upper stone. The lower stone was made from quartz sandstone and the upper stone was made from amphibolite [4]. The maximum length of the upper stone is 10.5 cm, maximum height is 4.8 cm [4], and its mass is 569 g. During saddle quern grinding, the upper stone is pushed back and forth against the lower stone with cereal between them. The rotary quern consists of a stationary lower stone and mobile upper stone, spindle, and handle. Both the lower and upper stone were made from granite [4]. During rotary quern grinding, the upper stone is revolved on the lower stone while grain is ground between them. The experimental grinding was done with dehusked Emmer wheat (*Triticum turgidum* subsp. *diccocum*). Emmer wheat is one of the founder crops, which were used since early agricultural periods [33]. Dimensions and material of the grinding tools as well as ground material were shown to influence grinding performance [34], therefore our results might be limited to the tools and ground material used in this study.

For both grinding tools, the volunteers were in a kneeling position (Fig 1C and 1D), which was used by Sládek et al. [4]. The kneeling position during saddle quern grinding is supported by lower limb bone alterations in past populations that used saddle querns or grinders [31,35,36], and by ethnographic observation [17]. Rotary querns may be placed on a platform for grinding in a standing position or on the ground for grinding in a kneeling or sitting position [37].

Before grinding data collection began, subjects practiced grinding with 1–5 trials, until they were able to execute the movement fluently and in required rhythm. Three experimental grinding tasks were performed: saddle quern grinding, clockwise rotary quern grinding, and anticlockwise rotary quern grinding, as shown in Fig 1E and 1F. Subjects ground on the saddle quern using both hands in a to-and-fro movement. Saddle quern grinding may also be performed using a circular movement, but the shape of the lower stone used in this study suggests a to-and-fro bimanual movement [4,38]. During saddle quern grinding, subjects were instructed to push down on the grain hard enough to break the seeds. The grain usually dispersed over the quern as subjects ground them, but subjects were instructed to continue the movement with the same pressure. To accommodate continual movement, no grain was added or removed by the subjects during the trial. Rotary quern grinding was performed using

**Fig 1. Grinding tools, grinding position, and grinding movements used in the experiment.** Saddle quern (A) and rotary quern (B) used in the experiment. Kneeling position used during experimental grinding (C, D). To-and-fro movement used during experimental saddle quern grinding (E) and clockwise and anticlockwise rotary movements used during rotary quern grinding (F).

the right hand in a circular motion in clockwise and anticlockwise directions. The same movements for saddle and rotary quern grinding were used previously by Sládek et al. [4]. Enough grain was added into rotary quern before each grinding trial to last for the whole trial. One cycle of saddle quern grinding was defined as one complete to-and-fro motion, and one cycle of rotary quern grinding was defined as complete revolution of the upper stone.

Grinding was performed in standardized tempo controlled by metronome, which was 175 and 85 bpm for saddle and rotary quern grinding, respectively. Each grinding cycle was performed at two metronome beats. This grinding tempo was previously used by Sládek et al. [4], where it was selected to be consistent with the preferences of subjects and to maximize grinding effectivity. The different tempo for saddle and rotary quern grinding shouldn't have an impact on the comparison of athletes and nonathletes, as both samples performed the grinding tasks in the same tempo but could have an impact on the comparison between the saddle and rotary quern grinding. When saddle quern grinding tempo was decreased to 110 bpm, maximum value of EMG signal increased by 1% and mean EMG value in the cycle increased by 3%. When clockwise rotary quern grinding tempo was increased to 110 bpm, maximum value of EMG signal increased by 34% and mean EMG value increased by 36%. Similarly, during anticlockwise rotary quern grinding maximum EMG value increased by 26% and mean EMG value increased by 38%. If we applied these alterations to previously published results [4],

change in tempo would increase the value of the relative difference between the saddle and rotary quern grinding. On the other hand, it wouldn't influence the pattern of the difference between the saddle and rotary quern grinding, which is a lower muscle activation during saddle than rotary quern grinding [4]. Grinding tasks performed in the present study are shown in S1–S3 Videos.

## Variables

To analyze the magnitude of muscle activation, the EMG curve of each muscle was analyzed using its maximum value (maxEMG, %MVC) and its mean value (meanEMG, %MVC), both of which represent percentage of EMG measured during maximum voluntary contraction. MaxEMG and meanEMG were used to compare muscle activation during grinding between athletes and nonathletes. To compare eight- and four-muscle model, the meanEMG values of individual muscles adjusted on physiological cross-sectional area (PCSA) (meanEMG$_{PCSA}$, % MVC s s$^{-1}$ cm$^2$) were calculated by multiplying meanEMG of each muscle by its PCSA. Comparison of eight- and four-muscle model was done only in nonathletes, therefore one set of PCSA values was used in the analysis. PCSA has been shown to reflect closely maximum force capacity of the muscle [13] and is standardly used in musculoskeletal modeling in biomedicine and sports medicine for estimation of maximum isometric force [e.g. 38,39]. In addition to approximation of force of each muscle, sum of meanEMG$_{PCSA}$ values (Summed meanEMG$_{PCSA}$, %MVC cm$^2$) was calculated to approximate the net force loading of the upper limb. The PCSA values were not measured in our sample but obtained from Holzbaur's study [40]. Since Holzbaur's study only provided PCSA of the whole deltoid and triceps muscle, PCSA values of deltoid and triceps parts were obtained using the proportions of each part to the whole muscle, which were adopted from the study of Langenderfer [41]. Since muscle PCSAs likely differ between athletes and nonathletes we avoided comparisons of Summed meanEMG$_{PCSA}$ between the samples. Summed meanEMG$_{PCSA}$ was used only within our sample of nonathletes for comparison of upper limb loading during saddle and rotary quern grinding between four- and eight-muscle model. Effectively, meanEMG$_{PCSA}$ is an approximate measure of the relative force exerted by the muscle over the course of its activity. Within each subject meanEMG$_{PCSA}$ values of all eight muscles were summed to obtain Summed meanEMG$_{PCSA}$ of the eight-muscle model, and meanEMG$_{PCSA}$ values of the anterior deltoid, infraspinatus, pectoralis major, and long head of the triceps brachii were summed to obtain Summed meanEMG$_{PCSA}$ of the four-muscle model. Summed meanEMG$_{PCSA}$ indicates the combined overall relative force level exerted by all of the included muscles during the measured activity.

To analyze coactivation between all analyzed muscles coactivation index was used [42]:

$$CI = \frac{2 \times (Int_1 + Int_2)}{Int_{tot}} \qquad (1)$$

Where Int$_1$ and Int$_2$ are mathematical integrals of smoothed EMG signals of two muscles. These integrals were computed in periods, in which the given muscle had lower values of EMG signal than the other muscle. Denominator Int$_{tot}$ is the sum of EMG integrals of both muscles in a cycle. Coactivation index has values between 0 and 1, with 0 denoting no overlap of EMG signal patterns of the muscles and 1 denoting perfect overlap of EMG signal patterns [42]. The index was computed for each pair of muscles in every subject.

## Statistical analysis

For the purpose of our first goal, we compared maxEMG and meanEMG in the sample of athletes and nonathletes using the Bonferroni post hoc test as recommended for data violating the

**Table 1. Maximum muscle activation (maxEMG, %MVC).**

| Muscle | Saddle | | Rotary: Clockwise | | Rotary: Anticlockwise | |
| --- | --- | --- | --- | --- | --- | --- |
| | Athletes (n = 10) | Nonathletes (n = 25) | Athletes (n = 10) | Nonathletes (n = 25) | Athletes (n = 10) | Nonathletes (n = 25) |
| Biceps b. | 4.2 (2.0–6.5) | 7.2 (5.8–8.6) | 17.5 (10.2–24.8) | 29.1 (24.5–33.7) | 53.4 (37.6–69.1) | 56.4 (46.4–66.3) |
| Anterior deltoid | 11.7 (5.8–17.6) | 12.8 (9.1–16.5) | 55.9 (43.8–68.0) | 59.3 (51.7–66.9) | 70.1 (54.3–85.9) | 85.1 (75.1–95.1) |
| Middle deltoid | 5.4 (3.0–7.9) | 8.3 (6.7–9.8) | 74.3 (61.4–87.2) | 73.7 (65.5–81.8) | 46.2 (32.5–59.8) | 51.4 (42.8–60.1) |
| Posterior deltoid | 12.4 (6.1–18.7) | 18.6 (14.5–22.6) | 90.3 (71.0–109.6) | 116.5 (104.3–128.7) | 55.9 (38.8–73.0)[a] | 84.2 (73.3–95.0)[a] |
| Infraspinatus | 11.0 (5.9–16.2) | 16.7 (13.5–20.0) | 68.7 (53.4–83.9) | 74.3 (64.7–83.9) | 58.1 (43.9–72.3) | 67.6 (58.6–76.6) |
| Pectoralis major | 11.2 (5.2–17.2) | 18.0 (14.2–21.8) | 59.9 (43.1–76.8) | 66.5 (55.9–77.2) | 82.6 (57.2–108.1) | 95.3 (79.2–111.4) |
| Triceps b. (lateral) | 34.8 (25.3–44.3) | 24.5 (18.5–30.5) | 55.3 (41.7–68.8) | 76.9 (68.3–85.5) | 36.9 (27.8–46.0) | 42.0 (36.3–47.8) |
| Triceps b. (long) | 35.1 (20.3–49.9) | 44.3 (35.0–53.7) | 69.2 (53.3–85.1) | 78.0 (68.0–88.1) | 30.7 (18.4–43.0) | 37.9 (30.2–45.7) |

Presented data are mean values with 95% confidence intervals in parentheses. All differences between athletes and nonathletes have p-values > 0.1, except for posterior deltoid during clockwise and anticlockwise rotary quern grinding. P-values are the results of the Bonferroni post hoc test. Biceps b., biceps brachii; triceps b., triceps brachii.

[a] Difference between athletes and nonathletes has p-value = 0.053.

assumption of sphericity [43]. Sphericity was analyzed using Mauchly's sphericity test. Bonferroni post hoc test was also used to compare coactivation indices between athletes and nonathletes. For our second goal, we compared Summed meanEMG$_{PCSA}$ between grinding types within the four- and eight-muscle models in the sample of nonathletes using the Tukey HSD test separately within each model. In meanEMG$_{PCSA}$ data the assumption of sphericity was met according to Mauchly's sphericity test. The significance threshold was set to 0.05. Post hoc tests were performed in Statistica for Windows (ver. 12, StatSoft, Inc., 1984–2013).

## Results

The values of maxEMG of athletes and nonathletes are shown in Table 1 and Fig 2. Mean maxEMG was lower in athletes than nonathletes in all muscles, except for the lateral head of the triceps brachii during saddle quern grinding and middle deltoid during clockwise rotary quern grinding. The differences are not significant at the .05 level. The highest difference on average was in posterior deltoid during anticlockwise rotary quern grinding, which athletes maximally activated to 55.9%MVC, while nonathletes maximally activated it to 84.2%MVC (p-value = 0.053). The second highest difference was in the posterior deltoid during clockwise rotary quern grinding, in which peak activation of athletes was 90.3%MVC, while in nonathletes it was 116.5%MVC (p-value = 0.174) lower mean maxEMG in athletes than nonathletes. On the contrary, the lowest difference between samples occurred in the anterior deltoid during saddle quern grinding, which was maximally activated to 11.7%MVC in athletes and 12.8% MVC in nonathletes. In both samples, the majority of muscles had on average lower maxEMG during saddle quern grinding than rotary quern grinding. An exception was the long head of the triceps brachii, which had higher maxEMG during saddle quern grinding than anticlockwise rotary quern grinding in both samples.

The values of meanEMG of athletes and nonathletes are shown in Table 2 and Fig 2. Athletes had similar meanEMG values nonathletes in all muscles except for the lateral head of the triceps brachii during saddle quern grinding. The highest mean difference, which was also the only significant one, occurred in the posterior deltoid during clockwise rotary quern grinding, in which meanEMG was 32.3%MVC in athletes and 44.6%MVC in nonathletes (p-value = 0.035). Differences between samples in other muscles were not significant. The lowest difference between samples was in the anterior deltoid during saddle quern grinding, which

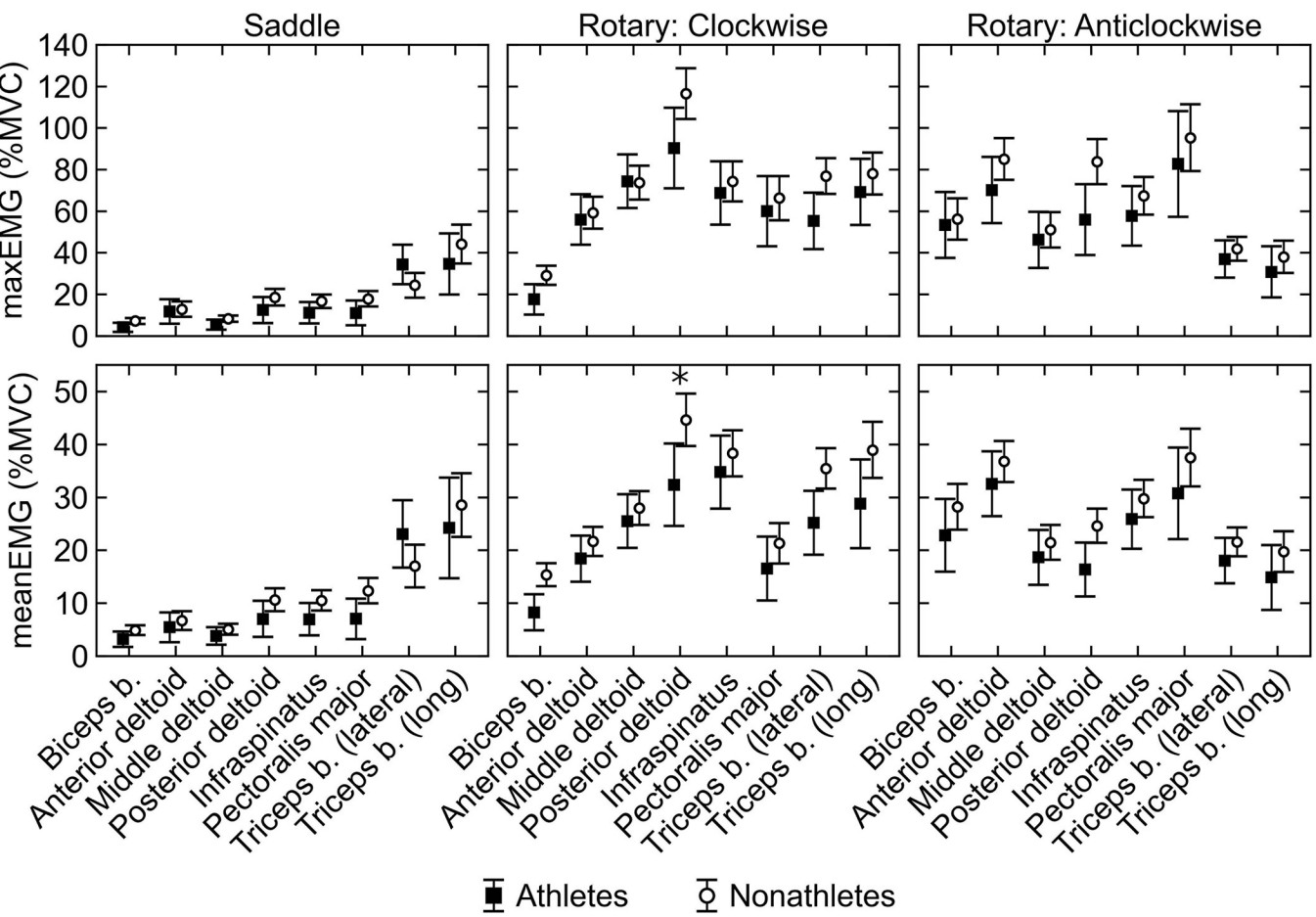

**Fig 2. Muscle activation during cereal grinding in athletes and nonathletes.** Maximum muscle activation (maxEMG, upper row) and mean muscle activity (meanEMG, lower row) in athletes (black squares) and nonathletes (white circles) during saddle quern (left column), clockwise rotary quern (middle column), and anticlockwise rotary quern (right column) grinding. Markers and whiskers indicate the means and 95% confidence intervals of the means, respectively. Significant difference between athletes and nonathletes (p < 0.05) is indicated by an asterisk. See Table 1 for abbreviations of muscles.

had meanEMG 5.4%MVC in athletes and 6.7%MVC in nonathletes. Saddle quern grinding had on average lower meanEMG than rotary quern grinding in the majority of muscles in both samples. An exception was the long head of the triceps brachii in both samples and lateral head of the triceps brachii in athletes, which had higher meanEMG during saddle quern grinding than anticlockwise rotary quern grinding.

The differences of coactivation index between athletes and nonathletes are shown in S3–S5 Tables. Higher antagonistic coactivation could indicate a less efficient movement strategy, during which the agonistic muscle must produce more force to overcome the moment of force of the antagonistic muscle. Coactivation indices were lower in athletes than nonathletes in 64 out of 84 comparisons. All of the differences between athletes and nonathletes were not significant, except for the coactivation between the lateral head of the triceps brachii and pectoralis major during saddle quern grinding, for which athletes had on average 30% lower (p-value = 0.048) coactivation index than nonathletes.

Summed meanEMG$_{PCSA}$ of the eight- and four-muscle models is shown in Table 3. By comparing Summed meanEMG$_{PCSA}$ of saddle and rotary quern grinding using the eight- and four-muscle models, we can show whether the larger set of muscles changes the estimates of

**Table 2. Mean muscle activation (meanEMG, %MVC).**

| Muscle | Saddle | | Rotary: Clockwise | | Rotary: Anticlockwise | |
|---|---|---|---|---|---|---|
| | Athletes (n = 10) | Nonathletes (n = 25) | Athletes (n = 10) | Nonathletes (n = 25) | Athletes (n = 10) | Nonathletes (n = 25) |
| Biceps b. | 3.2 (1.7–4.6) | 4.9 (4.0–5.8) | 8.3 (4.8–11.7) | 15.4 (13.2–17.5) | 22.8 (15.9–29.7) | 28.2 (23.9–32.5) |
| Anterior deltoid | 5.4 (2.6–8.2) | 6.7 (4.9–8.5) | 18.4 (14.0–22.8) | 21.7 (18.9–24.4) | 32.5 (26.4–38.7) | 36.8 (32.9–40.6) |
| Middle deltoid | 3.8 (2.1–5.4) | 5.1 (4.0–6.1) | 25.5 (20.5–30.5) | 28.0 (24.8–31.1) | 18.6 (13.5–23.8) | 21.5 (18.2–24.7) |
| Posterior deltoid | 7.0 (3.6–10.4) | 10.6 (8.5–12.8) | 32.3 (24.6–40.1)[a] | 44.6 (39.7–49.6)[a] | 16.4 (11.3–21.5) | 24.6 (21.4–27.8) |
| Infraspinatus | 6.9 (3.9–10.0) | 10.5 (8.6–12.4) | 34.8 (27.9–41.7) | 38.3 (33.9–42.7) | 25.9 (20.3–31.5) | 29.8 (26.2–33.3) |
| Pectoralis major | 7.0 (3.2–10.8) | 12.3 (9.9–14.7) | 16.5 (10.5–22.6) | 21.3 (17.5–25.1) | 30.7 (22.1–39.4) | 37.5 (32.0–43.0) |
| Triceps b. (lateral) | 23.1 (16.7–29.4) | 17.0 (13.0–21.0) | 25.2 (19.1–31.2) | 35.4 (31.6–39.3) | 18.0 (13.7–22.3) | 21.6 (18.8–24.3) |
| Triceps b. (long) | 24.2 (14.7–33.7) | 28.5 (22.5–34.5) | 28.8 (20.4–37.1) | 38.9 (33.6–44.2) | 14.8 (8.7–21.0) | 19.7 (15.9–23.6) |

Presented data are mean values with 95% confidence intervals in parentheses. All differences between athletes and nonathletes have p-values > 0.1, except for posterior deltoid during clockwise rotary quern grinding. P-values are the results of the Bonferroni post hoc test. See Table 1 for abbreviations of muscles.

[a] Difference between athletes and nonathletes has p-value = 0.035.

the upper limb loading during saddle and rotary quern grinding. Saddle quern grinding had on average significantly lower Summed meanEMG$_{PCSA}$ than rotary quern grinding (in either direction) in both models. Summed meanEMG$_{PCSA}$ during saddle quern grinding was on average 2.7 times lower than that during clockwise rotary quern grinding (p-value = 0.0001) in the eight-muscle model and 2.3 times lower (p-value = 0.0001) in the four-muscle model. Summed meanEMG$_{PCSA}$ during saddle quern grinding was on average 2.6 times lower than that during anticlockwise rotary quern grinding (p-value = 0.0001) in the eight-muscle model and 2.5 times lower (p-value = 0.0001) in the four-muscle model. In the four-muscle model, Summed meanEMG$_{PCSA}$ during clockwise rotary quern grinding was non-significantly lower than that during anticlockwise rotary quern grinding (p-value = 0.066). On the contrary, in the eight-muscle model, Summed meanEMG$_{PCSA}$ during clockwise rotary quern grinding was non-significantly higher than that during anticlockwise rotary quern grinding (p-value = 0.354).

## Discussion

Our first goal was to compare muscle activation during cereal grinding between athletes and nonathletes. We expected athletes to have lower muscle activation than nonathletes during grinding due to lower antagonistic coactivation in athletes. While there was a trend of lower activation in athletes than in nonathletes, the only significant difference occurred in posterior deltoid during clockwise rotary quern grinding, where the activation was significantly lower in athletes compared to nonathletes. Coactivation was lower in athletes than nonathletes in most

**Table 3. Total activity of all measured muscles adjusted to PCSA (Summed meanEMG$_{PCSA}$, %MVC cm$^2$) in the eight- and four-muscle models in nonathletes.**

| Type of grinding | Eight-muscle model (n = 25) | Four-muscle model (n = 25) |
|---|---|---|
| Saddle | 1168.4 (995.7–1341.0)[a,b] | 747.8 (624.8–870.9)[a,b] |
| Rotary: Clockwise | 2727.8 (2524.5–2931.1)[b,c] | 1474.4 (1338.5–1610.3)[c] |
| Rotary: Anticlockwise | 2441.7 (2203.8–2679.6)[a,c] | 1495.3 (1342.2–1648.3)[c] |

Presented data are mean values with 95% confidence intervals in parentheses. Significance was determined using the Tukey HSD test.

[a] Significantly different from clockwise rotary quern grinding.

[b] Significantly different from anticlockwise rotary quern grinding.

[c] Significantly different from saddle quern grinding.

muscle pairs, but significant difference occurred only in one muscle pair, which was not antagonistic. Our results therefore suggest that nonathletes can be used for reconstruction of saddle and rotary quern grinding. Furthermore, the broader test of the relative muscle activity in saddle versus rotary quern grinding is fully consistent; in both samples rotary quern grinding involves significantly more muscle activation than saddle quern grinding. Our second goal was to test the influence of the number of analyzed muscles on the estimate of upper limb loading during cereal grinding. Again, both the eight- and four-muscle models indicated that the upper limb is loaded significantly less during saddle quern grinding than rotary quern grinding, which suggests that the upper limb muscles may be reduced to the four-muscle model for comparison of saddle and rotary quern grinding.

## Muscle activation during grinding in athletes and nonathletes

There was a trend of lower average maxEMG and meanEMG in athletes than nonathletes, in which athletes had on average 16% lower maxEMG and 19% lower meanEMG than nonathletes during cereal grinding. On the other hand, significant differences between athletes and nonathletes occurred only in the posterior deltoid during clockwise rotary quern grinding. Furthermore, both athletes and nonathletes had lower activation during saddle quern grinding than rotary quern grinding in the majority of muscles, supporting the previous analysis based on nonathletes.

Lower activation of the upper limb muscles in athletes than nonathletes could be caused by lower antagonistic coactivation, higher muscle activation in other parts of the body, or higher muscle hypertrophy in athletes than nonathletes. Antagonistic coactivation would have an impact on both muscle activation and force because with lower coactivation, the agonistic muscle would have to produce less force to overcome the force of the antagonistic muscle. Even though in most muscle pairs coactivation was on average lower in athletes than nonathletes, no significant difference was found between samples in the coactivation of the antagonistic muscles. The lack of significant differences in coactivation between athletes and nonathletes could be also associated with similar timing of muscle activation in both samples (Fig 3). In any case, the similar levels of antagonistic coactivation in athletes and nonathletes support the use of nonathletes in experimental cereal grinding.

Another possibility is that lower activation of the upper limb muscles in athletes than nonathletes was caused by the activation of the muscles of other parts of the body than the upper limbs. For example, it was observed that Hopi women used the whole body to produce rhythmic movements during grinding on saddle quern [44]. Rowers may be able to activate the muscles of the whole body better than nonathletes because during rowing they generate power with the upper and lower limbs and torso [45]. Therefore, we cannot rule out the possibility that lower activation of the upper limb muscles in athletes was caused by their higher activation of the muscles in other parts of the body. However, again it is notable that differences between athletes and nonathletes are largely insignificant, with the exception of a single muscle (posterior deltoid), and the direction of contrast between saddle and quern grinding was consistent between the two samples.

Alternatively, non-significantly lower activation of the upper limb muscles in athletes than nonathletes may be caused by increased muscle hypertrophy in athletes, indicating different muscle force between athletes and nonathletes [46,47]. It was observed that with increasing muscle hypertrophy, activation per muscle force decreases [47]. Therefore, a hypertrophied muscle may produce the same muscle force with lower muscle activation than a muscle without hypertrophy. Muscle hypertrophy can be increased by both resistance training [48] and endurance training [49]. Therefore, it may be expected in the upper limb muscles of rowers.

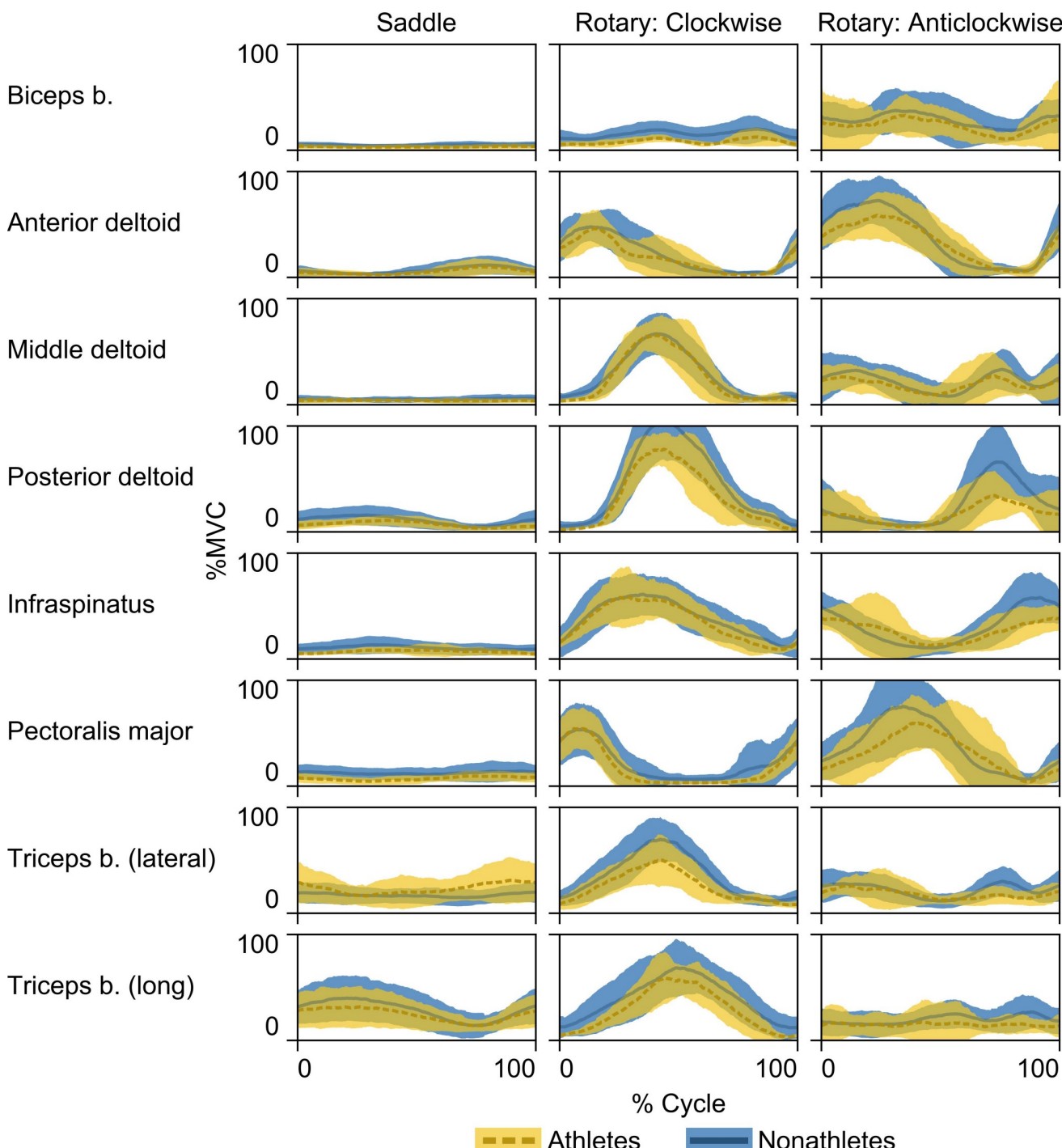

**Fig 3. Average EMG curves.** Results are shown for athletes (yellow, dashed line) and nonathletes (blue, solid line) during saddle quern (left column), clockwise rotary quern (middle column), and anticlockwise rotary quern (right column) grinding. Darker lines and colored areas indicate the means and ± standard deviations, respectively. See Table 1 for abbreviations of muscles.

Rowing power output was shown to be correlated with hypertrophy of the muscles that extend the elbow [45], which suggests that those muscles could be hypertrophied in our athletic sample as well. The relationship between muscle activation and contraction force is further

influenced by the length and the pennation angle of contracting muscle fibers [15], and type of muscle fibers [50]. Nevertheless, we cannot rule out the possibility that lower muscle activation in athletes was offset by hypertrophied muscles in athletes.

Posterior deltoid was the only muscle with significantly lower mean activation in athletes than nonathletes during clockwise rotary quern grinding. Posterior deltoid has origin on scapular spine and inserts on deltoid tuberosity and its function is shoulder extension [51]. Rowing movement contains shoulder extension over an angular range of 110° [52] and maximum activation of posterior deltoid is on average 88% MVC during horizontal rowing movements [53]. Repeated shoulder extension and activation of posterior deltoid might cause hypertrophy of posterior deltoid, lower coactivation of posterior deltoid with anterior deltoid, pectoralis major, and biceps brachii, or better coordination of shoulder extension. The decrease of posterior deltoid activation could exist in early agricultural females too, as posterior deltoid was one of the two most active muscles during clockwise rotary quern grinding in both samples. Our results showed no significant difference between athletes and nonathletes in coactivation of posterior deltoid with either anterior deltoid, pectoralis major, or biceps brachii, therefore decreased muscle coactivation would not explain the lower posterior deltoid activation in our athletes than nonathletes. The difference could be caused by different coordination with other muscles (e.g., higher activation of back extensors) in athletes than nonathletes. In such case, less functional adaptation of humerus to rotary quern grinding might take place at deltoid tuberosity than would be suggested by the results of nonathletes. On the other hand, posterior deltoid of athletes might have been hypertrophied, in which case the relationship between muscle activation and muscle force would not be so straightforward. In such case the lower activation of posterior deltoid in athletes than nonathletes would not imply less functional adaptation at the deltoid muscle insertion site than suggested by nonathletes.

Our results might suggest that influence of athletic samples is limited to reconstructions of tasks in which the athletes are specialized, such as reconstruction of spear throwing using javelin throwers [6]. Furthermore, the difference between athletes and nonathletes might present itself only in some variables describing performance, such as success rate during spear throwing [6], while other measures, such as muscle activation, might be similar in athletes and nonathletes. Future studies of cereal grinding might therefore compare athletes with nonathletes in other measures, such as grinding efficiency.

### Influence of analyzed muscles on the estimated upper limb loading

Approximation of muscle force on upper limb (summed meanEMG$_{PCSA}$) was significantly lower during saddle quern grinding than rotary quern grinding in both eight- and four-muscle models. The similarity of results from both models for comparison of saddle and rotary quern grinding supports reducing the upper limb muscles to the four-muscle model.

In our results, both eight- and four-muscle models indicate that less muscle force during saddle quern grinding than rotary quern grinding. However, the eight- and four-muscle models differed in comparison of produced muscle force during clockwise rotary quern grinding and the other two grinding types. The eight-muscle model suggested a higher difference in muscle force between clockwise rotary and saddle quern grinding than the four-muscle model. The observed difference between the eight- and four-muscle models may be explained by the muscles included in the models and their importance for either grinding type. During saddle quern grinding, the highest meanEMG$_{PCSA}$ was in the pectoralis major and long head of the triceps brachii, which were included in both models. During anticlockwise rotary quern grinding, the highest meanEMG$_{PCSA}$ was in the anterior deltoid, infraspinatus, and pectoralis major, which were also included in both models. The inclusion of these muscles in both

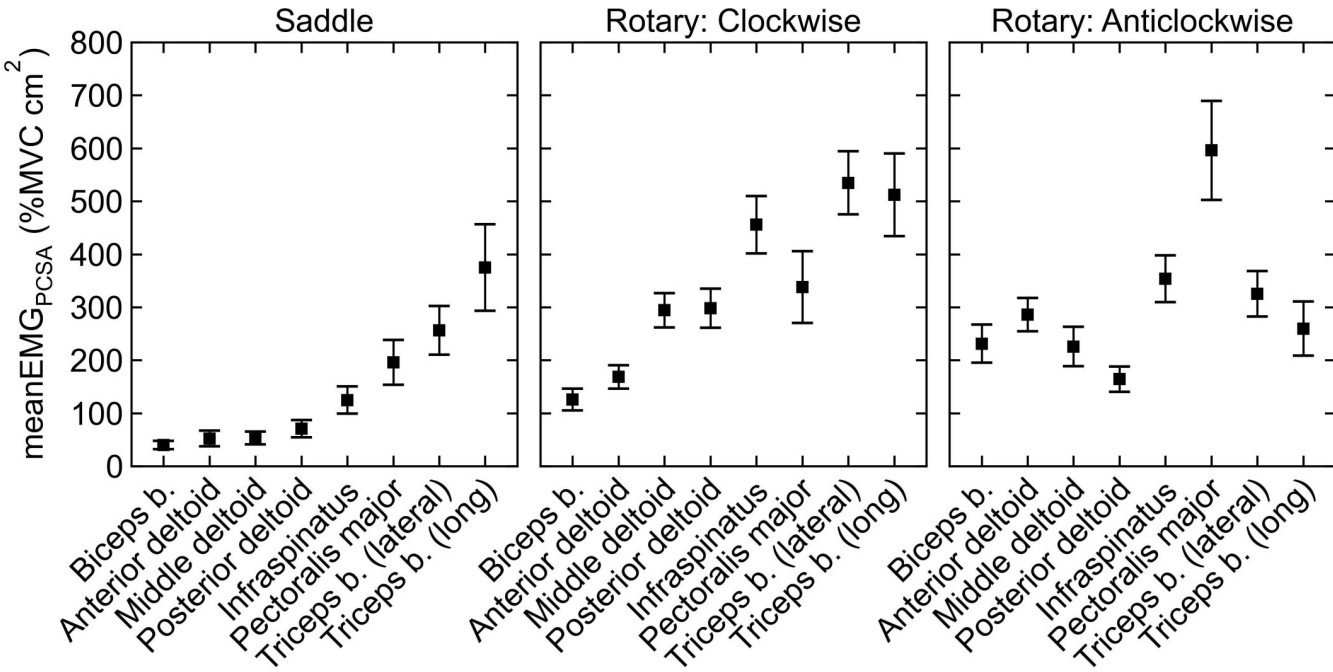

**Fig 4. Mean muscle activity adjusted for PCSA of the muscles (meanEMG$_{PCSA}$) in nonathletes.** Results are shown for saddle quern (left), clockwise rotary quern (middle), and anticlockwise rotary quern (right) grinding. Markers and whiskers indicate the means and 95% confidence intervals of the means, respectively. See Table 1 for abbreviations of muscles.

models perhaps led to a similar ratio of Summed meanEMG$_{PCSA}$ of anticlockwise rotary and saddle quern grinding in both models. During clockwise rotary quern grinding, the second and third highest meanEMG$_{PCSA}$ were in the posterior and middle deltoid, respectively. As these two muscles were only in the eight-muscle model, their inclusion may have increased the ratio between Summed meanEMG$_{PCSA}$ of clockwise rotary and saddle quern grinding in the eight-muscle model. Therefore, the four-muscle model may underestimate the muscle force produced during clockwise rotary quern grinding relative to saddle quern grinding. Nevertheless, both models showed significantly lower Summed meanEMG$_{PCSA}$ in saddle quern grinding than rotary quern grinding. Therefore, the upper limb muscles could be reduced to four muscles for comparison of upper limb muscle force during saddle and rotary quern grinding.

In our results, the most active muscle during saddle quern grinding was the long head of the triceps brachii, which had the highest values of activation (maxEMG and meanEMG) and the second highest activation adjusted to PCSA (meanEMG$_{PCSA}$; Fig 4). The long head of the triceps brachii was also the only muscle that had higher maxEMG and meanEMG during saddle quern grinding than rotary quern grinding. High values of maxEMG and meanEMG of the long head of the triceps brachii were also observed in a previous cereal grinding study [4]. Primary function of triceps brachii is elbow extension which is performed when pushing the upper stone forward on the saddle quern. Triceps brachii force might be also used to stabilize moment of force in the direction of elbow flexion caused by subjects pressing down on the saddle quern. High activation of the triceps brachii relative to other muscles during saddle quern grinding shows the possibility of using its origin and insertion in future studies of entheseal changes associated with saddle quern grinding. Saddle quern grinding has previously been suggested to cause entheseal changes at the deltoid insertion [28]. In our results, the values of meanEMG$_{PCSA}$ of each deltoid part were on average lower than those of the most active muscles, but the combined value of all deltoid parts would be the second highest of the analyzed

muscles. Saddle quern grinding was also previously suggested to be the cause of prominent insertion of the biceps brachii [28]. Therefore, it would be expected to have high values of activation. On the contrary, our results showed that the biceps brachii had the lowest maxEMG, meanEMG, and meanEMG$_{PCSA}$ of measured muscles during saddle quern grinding. During rotary quern grinding, the infraspinatus and pectoralis major had the highest meanEMG$_{PCSA}$; thus, their insertions could be candidates for entheseal changes associated with rotary quern grinding. Nevertheless, if meanEMG$_{PCSA}$ values of the anterior, middle, and posterior deltoids are combined, the resulting value is higher than other muscles' meanEMG$_{PCSA}$ during rotary quern grinding; thus, the deltoid may be significant for rotary quern grinding as well. For further research of physical activity during the adoption of agriculture, the pectoralis major, triceps brachii, and previously suggested deltoid may be candidate muscles for analysis of entheseal changes. For studies of physical activity during the intensification of agriculture, the deltoid muscles, infraspinatus, and pectoralis major may be candidate muscles for analysis of entheseal changes.

## Conclusions

Our results showed a trend of lower activation in the muscles of athletes relative to nonathletes; however, these differences mostly were not significant. Furthermore, both athletes and nonathletes had lower activation during saddle quern grinding than rotary quern grinding in the majority of muscles. Our results therefore indicate that both athletes and nonathletes can be used for comparison of upper limb loading during saddle and rotary quern grinding. Our results also suggest that a homogenous sample would be favorable for estimation of muscle force during cereal grinding to minimize influence of subjects' muscle hypertrophy on muscle activation.

Both eight- and four-muscle models showed lower upper limb loading during saddle quern grinding than rotary quern grinding, which suggests that the upper limb muscles may be reduced to the previously used four-muscle model for comparison of saddle and rotary quern grinding.

Finally, our results also suggest refinements in the use of osteological markers of entheseal changes to analyze physical activity changes during the adoption of agriculture. These changes associated with grinding should be observed for the deltoid, infraspinatus, and pectoralis major for both saddle and rotary quern grinding or for the long head of the triceps brachii for saddle quern grinding. However, the low level of biceps brachii involvement raises questions about the interpretation of prominent radial tuberosity development as specifically related to saddle quern grinding [28] and suggests that alternative strenuous activities might be considered as responsible for this skeletal indicator.

More generally, our results confirm that experimental replicative research on activity patterns involving modern populations may yield fairly robust results, despite the contrast in level of conditioning of most modern populations, and with a judiciously chosen, limited set of monitored muscles. Furthermore, it offers the possibility of more broadly examining muscle activation in a range of prehistoric activities to better assess and test interpretations of prehistoric skeletal modifications.

## Supporting information

**S1 Fig. Placement of EMG sensors.** Images were generated from www.biodigital.com and edited afterwards.
(TIF)

**S1 Table. Maximum voluntary contraction (MVC) values.**
(DOCX)

**S2 Table. Origin, insertion, function, and MVC tests of muscles used in the study.**
(DOCX)

**S3 Table. Coactivation indices during saddle quern rotary quern grinding in athletes and nonathletes.**
(DOCX)

**S4 Table. Coactivation indices during clockwise rotary quern grinding in athletes and non-athletes.**
(DOCX)

**S5 Table. Coactivation indices during anticlockwise rotary quern grinding in athletes and nonathletes.**
(DOCX)

**S1 Video. Cereal grinding using saddle quern.** Synchronized acquisition of electromyography data is shown in the right part of the video. Electromyography data was full wave rectified and smoothed using root mean square function. See Table 1 for abbreviations of muscles.
(MP4)

**S2 Video. Cereal grinding on rotary quern in clockwise direction.** Synchronized acquisition of electromyography data is shown in the right part of the video. Electromyography data was full wave rectified and smoothed using root mean square function. See Table 1 for abbreviations of muscles.
(MP4)

**S3 Video. Cereal grinding on rotary quern in anticlockwise direction.** Synchronized acquisition of electromyography data is shown in the right part of the video. Electromyography data was full wave rectified and smoothed using root mean square function. See Table 1 for abbreviations of muscles.
(MP4)

## Acknowledgments

We would like to thank the participants from Charles University and from the rowing teams Český veslařský klub Praha, Veslařský klub Smíchov, and SK HAMR. We also wish to thank the coaches and staff of the participating rowing teams. We would like to thank Iva Brynychová, Pavla Alexia Jarešová, Klára Kosatíková, and Zuzana Matějovská for their help with data collection. We would also like to thank Jakub Otáhal for the advice about EMG data collection.

## Author Contributions

**Conceptualization:** Michal Struška, Martin Hora, Vladimír Sládek.

**Formal analysis:** Michal Struška.

**Funding acquisition:** Martin Hora.

**Investigation:** Michal Struška.

**Methodology:** Michal Struška, Martin Hora, Vladimír Sládek.

**Project administration:** Michal Struška.

**Resources:** Vladimír Sládek.

**Supervision:** Vladimír Sládek.

**Visualization:** Michal Struška, Martin Hora, Thomas R. Rocek, Vladimír Sládek.

**Writing – original draft:** Michal Struška.

**Writing – review & editing:** Michal Struška, Martin Hora, Thomas R. Rocek, Vladimír Sládek.

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
