## [Decision Letter · Decision Letter 0]

26 Mar 2021

PONE-D-20-36922

Influence of upper limb training and analyzed muscles on estimate of physical activity during cereal grinding using saddle quern and rotary quern

PLOS ONE

Dear Dr. Struška,

Thank you for submitting your manuscript to PLOS ONE. After careful consideration, we feel that it has merit but does not fully meet PLOS ONE’s publication criteria as it currently stands. Therefore, we invite you to submit a revised version of the manuscript that addresses the points raised during the review process.

Reviewers 2 and 3 had relatively minor comments to consider, and I believe that these will be straightforward to address. However, Reviewer 1 is critical of the study, with a series of comments and recommendations. Most serious concern questions about the validity of the PCSA estimates, which needs to be addressed in a resubmission and rebuttal. 

We look forward to receiving your revised manuscript.

Kind regards,

Michael D. Petraglia, Ph.D.

Academic Editor

PLOS ONE

Journal Requirements:

1. Please ensure that your manuscript meets PLOS ONE's style requirements, including those for file naming. The PLOS ONE style templates can be found athttps://journals.plos.org/plosone/s/file?id=wjVg/PLOSOne_formatting_sample_main_body.pdf andhttps://journals.plos.org/plosone/s/file?id=ba62/PLOSOne_formatting_sample_title_authors_affiliations.pdf

Additional Editor Comments (if provided):

Reviewers' comments:

Reviewer's Responses to Questions

**Comments to the Author**

1. Is the manuscript technically sound, and do the data support the conclusions?

Reviewer #1: Partly

Reviewer #2: Yes

Reviewer #3: Yes

2. Has the statistical analysis been performed appropriately and rigorously? 

Reviewer #1: Yes

Reviewer #2: I Don't Know

Reviewer #3: Yes

3. Have the authors made all data underlying the findings in their manuscript fully available?

Reviewer #1: Yes

Reviewer #2: Yes

Reviewer #3: Yes

4. Is the manuscript presented in an intelligible fashion and written in standard English?

Reviewer #1: Yes

Reviewer #2: Yes

Reviewer #3: Yes

5. Review Comments to the Author

Reviewer #1: Please see my more detailed review comments attached, but I do have a concern with the PCSA estimates that were used and feel they are inappropriate for the study participants. PCSA or a proxy for it should have been built into the study design, or more appropriate reference data chosen from the literature.

Reviewer #2: The authors compare athletes vs nonathletes in two types of experimental grinding activities that simulate technologies commonly used in ancient societies. A goal of the study was to evaluate the efficacy of using nonathletes in experimental work as previous work had suggested athletes were necessary. The found that there was little difference in muscle activation between the two groups and suggests that nonathletes could be used. The study also more precisely identified what groups are activated during particular grinding activities. This is useful as it will assist in identifying which muscle insertions can be examined in ancient skeletal remains to reconstruct past activities. The methods and data that support these claims are presented and analyzed although I wasn't familiar with some of the statistics used.

In watching the cool videos of the grinding experiments I noticed the upper stone used with the saddle quern seemed unusually small. Typically these stones are about the width of the lower stone. The smaller stone would require a lot less force which could impact the utility of the experiments. Also while the text says grain was used in the experiments there is no further discussion of that. The grain on the grinding surfaces has a direct effect on the friction and thus the force required to move the stone. There is also no discussion of the manipulation of the grain. Grinders have to constantly stop and add and remove grain and would affect the experimental rhythm. Ideally the authors may have measured the work or forces involved in the movement of the machines, if so these data should be included. Otherwise they should at least provide the weight and dimensions of the upper stone and discuss the grain used in the experiment and its manipulation. Also I didn't see where they identified the material type of the stones. Actually grinding grain requires substantially more work that simply simulating the movement by moving the stone back and forth and would or course effect muscle activation. I am less familiar with rotary querns but some of these issues would apply, certainly the friction between the two grinding faces is an important part of the physics. Addressing these issues would foreclose concerns with these experiments.

Reviewer #3: This is an interesting study. My only comment is with regard to the age and body size differences between the athletes and non-athletes. The athletes are younger, taller, and heavier. Would this not indicate they may not require as much muscle activation to produce the same force? However, I don't think it would have any impact on the results of your study.

6. PLOS authors have the option to publish the peer review history of their article (what does this mean?). If published, this will include your full peer review and any attached files.

Reviewer #1: No

Reviewer #2: **Yes: **Robert J Hard

Reviewer #3: No

---

## [Author Response · Author response to Decision Letter 0]

1 Jul 2021

Editors

PLOS ONE

July, 2021

Dear Dr. Petraglia,

thank you and the reviewers for a constructive criticism of our manuscript and for the opportunity to revise and resubmit. We are submitting our revised manuscript with the title: Influence of upper limb training and analyzed muscles on estimate of physical activity during cereal grinding using saddle quern and rotary quern. In our response we use references to pages and lines, which correspond to pages and lines in the file “Revised Manuscript with Track Changes”.

On behalf of my co-authors, I thank you for your consideration of this resubmission. We appreciate your time and look forward to your response.

Sincerely

Michal Struška

Reviewer #1:

1) The authors need to define muscle activation and what affects it. They also need to provide much more detail on what the upper limb movements involved in saddle and rotary quern grinding are, and what muscles are involved in these movements. A table with the relevant muscles, their origin and insertion points, and their primary movement would be useful. This would apply to some of the more specific areas below as well:

• page 5, lines 87-89: define muscle activation, what does it mean “could have activated their muscles less”?

• page 6, lines 107-116

• page 6, the first study goal

• all of page 8 in the methods (Assessment of muscle activation)-- why were these muscles chosen? What movements do they do, and where do they originate and insert?

• discussion section pages 19-20- a lot of time is spent on the lower activation in athletes than controls, so muscle activation and the factors that cause variation in it should be clearly set out in the introduction

As suggested, we added a paragraph about muscle activation, its relationship with muscle force, and factors that influence this relationship (page 5, lines 83-90). We also added a paragraph describing movements involved in saddle quern and rotary quern grinding (page 8, lines 168-174) and table describing the origin, insertion, and function of the muscles selected for our study (S2 Table).

2) The authors should also include a brief description of bone functional adaptation in response to muscle force, and the ‘muscle-bone’ unit.

• page 5, line 90: “skeletal adjustments”, be more specific

Following the suggestion, we added description of the process of bone functional adaptation to muscle force on page 5, lines 83-86.

3) More detail is needed about how the maximum voluntary contraction data were obtained for each muscle. There is currently on a small section on this at the end of page 8. How were MVCs obtained, was this done by dynamometry? How many sets/reps etc? When were these measured- immediately before the quern trials? Did the athletes have higher MVC than the non-athletes? A summary table with the anthropometry, age, physical activity, and MVC data for the two groups (athletes, controls) would be beneficial to visually compare.

We added a paragraph detailing the MVC tests (page 9-10, lines 194-203) and we added a table containing mean MVC values (S1 Table). We omitted the anthropometry, age, and physical activity from the table, because we did not want to repeat information which is already present in the text.

4) Did the volunteers get any practice trials of the grinding before the recordings were started? If so, how many?

We added information on practice trials on page 11, on lines 230-231.

5) Were the subjects grinding actual grains during the trials, or were they just performing the movements without any grains present? Either way, how would this affect the muscle effort required? Presumably for a saddle quern, if you are actually trying to grind grains down you would be pressing down with more effort than if you are just moving the stone back and forth for no reason, and then also choice of grain would likely matter for how tough they are to grind down. More information here would be useful.

As suggested by both reviewer 1 and reviewer 2, we added description of ground material (page 10, lines 214-216). The absolute lack of grain during grinding on saddle quern would probably lead to lower pressure on the upper stone and therefore lower muscle activation, as subjects would probably choose lower pressure and would not have any feedback about their grinding effort. We instructed subjects to apply enough pressure to break the grain and to continue with the same pressure even if the grain dispersed over the lower stone. We added this description on page 11, lines 236-240. From our experience, when there is no grain in the rotary quern, grinding movement becomes more difficult, therefore it would probably lead to higher muscle activation by subjects. We added enough grain to rotary quern to last for the whole trial, so there was no grinding without grain. We added this information on page 11, line 243.

6) Choice of grinding tempo: Can you provide more information on why these grinding tempos were used? These are different subjects than in the cited Sládek paper, so should subject preference have been re-tested here? What would happen if subjects were just allowed to grind at their preferred tempo? It is stated that the different tempo for saddle and rotary quern grinding shouldn’t impact differences between athletes and non-athletes, but I am not sure that that is true, as it seems as though weaker individuals may need the momentum of a faster pace to get tough grains ground down. That may be wrong, but if you can justify these choices and back them up, that would be helpful.

The tempo was selected in the Sládek et al. paper to be consistent with preferences of the subjects. If subjects were given choice of preferred tempo, it would lead to different velocity of contraction among subjects. This would further complicate the relationship between muscle activation and force, due to the muscle force-velocity relationship, from which we can infer that subjects with higher contraction velocity would produce lower amount of force per given activation. Faster tempo might lead to better efficiency of weaker subjects, but our aim was only to compare muscle activation between athletes and nonathletes during the same movement.

7) I found it difficult to understand what iEMG is actually measuring, is this an integrated EMG analysis (signals integrated for the whole contraction to provide work per contraction)? It looks like your iEMG is a value per cycle, but can you provide more detail and clarification explaining what this is? It would also be useful to more explicitly explain that maxEMG (in %MVC) is representing the percentage of its maximal voluntary contraction force that that muscle is using. This is also why a better explanation of how MVC was obtained for each muscle is necessary as previously mentioned.

Following reviewer’s comment, we replaced iEMG with mean EMG value of the cycle, which is easier to understand. The mean of EMG has the same value as iEMG, due to the way iEMG was calculated. We added the suggested explanation of both maxEMG and newly added meanEMG on page 12, on lines 265-268.

8) I have serious reservations with the PCSA values used to adjust iEMG values (page 11). These were not measured from the participants themselves, but were taken from a study that got them from just TWO individual female cadavers, one of whom was 91 years old and the other 28. Were the 91-year old’s muscle cross-sectional areas used? If so, these cannot reasonably be considered representative of young women with an average age of 16 years old in this study. Further, the PCSAs of the relevant muscles would be expected to vary substantially between the athlete and the non-athlete group, and it would seem to me that some measure of muscle area or a proxy for it should have been incorporated into the study design.

My concerns with the iEMGPCSA are large enough that I think this variable should be removed altogether, along with the summed iEMGPCSA data, unless PCSA estimates can be obtained from a larger comparative dataset closer in age and physical activity level to the subjects. There should be muscle cross-sectional area data from pQCT, CT, and/or MRI for at least some of the relevant muscles published in the sport sciences literature. Otherwise, a proxy for PCSA could be used, like upper arm circumference, if this was obtained.

We replaced PCSA values from Langenderfer et al. (2004) with PCSA values from the study of Holzbaur et al. (2007). Values in the study of Holzbaur et al. are based on larger sample of 10 living subjects, whose age was between 24 and 37 years. Values from the study of Langenderfer et al. are now used only to calculate PCSA of parts of deltoid and triceps muscle, which were measured only as a whole in the study of Holzbaur et al. Values of PCSA were used only for the comparison of eight- and four-muscle model in nonathletes and not for comparisons between samples, which we stress out in the revised methods (page:13, lines 272-274 and 282-286).

9) In a related note, page 20-21 of the discussion section spend a lot of time discussion the possible effect of muscle hypertrophy in the athletes relative to nonathletes as a reason behind the trend towards lower activation in the former relative to latter (though largely non-significantly so). This seems to then clearly suggest that using the same estimates of PCSA for both groups of females is not a good idea, as the authors themselves suggest that it is likely the athletes would have had relatively larger muscles.

Values of PCSA were used only in comparison of eight- and four-muscle model, which was done only in nonathletes. We clarified this by adding explicit description of our use of the PCSA values on page 13, on lines 272-274 and 282-286.

10) Given the very few significant differences, the authors spend a lot of time describing trends in the data that are not actually significant. Clearly these non-significant results provide good support for the use of experimental muscle data from control subjects as a comparative reference sample for pre-industrial behaviours, but the authors could focus more of their discussion on the main significant finding: that less active females seem to have to recruit their posterior deltoid more than more active females when doing rotary quern grinding (maxEMG not significant but close to it and supports the significant iEMG finding). The authors should explore why this might be:

• where does this muscle originate and insert?

• what does it do to move the shoulder/elbow?

• why might it be more recruited in control subjects?

• Is this muscle used in the rowing movement?

• How might this affect interpretations of behaviour in the past from midshaft humeral cross-sections taken at/near the deltoid insertion? etc

Related to this, on pages 19-20: “Lower activation of the upper limb muscles in our athletic than nonathletic sample”—however all of the maxEMG values were non-significant and all but one of the iEMG values were non-significant, so this is a bit of a grand statement used in this and the following paragraphs. I think the focus should be primarily on the posterior deltoid iEMG specifically, with a shorter mention of the general trend towards lower mean values in athletes but clear articulation that these were not significant, and perhaps a larger sample size of athletes was required.

We added a paragraph discussing the origin and insertion of posterior deltoid, its role in rowing, and its significantly lower activation during clockwise rotary quern grinding in athletes than nonathletes (page 23-24, lines 453-474).

Other miscellaneous comments:

page 4, line 79: “we don’t know what influence an athletic sample would have on the previous experimental reconstruction of other tasks, such as spear thrusting [3]”

-at least some of the individuals in this study cited were trained athletes (gymnasts), though I don’t think this is specified in the original paper

We deleted the statement about spear thrusting.

page 5, line 83: reword this to specify western industrialized females— not ALL women have no experience with grain processing, many women still do this in some parts of the world, as you mention on page 5 line 93

We changed the statement as suggested.

page 5, line 93: are saddle and rotary querns always used concurrently in early Holocene populations? When do these technologies first appear?

Saddle and rotary quern are not used concurrently, as saddle querns became ubiquitous in the Neolithic, while rotary querns first appear in archaeological records in the Iron Age. With our statement we meant to say, that we cannot find contemporary females skilled in both saddle and rotary quern grinding, which would be the ideal case for comparison of saddle quern and rotary quern grinding. We tried to clarify this on page 5-6, lines 104-106.

page 5, last paragraph: some precedent here could be Macintosh and colleagues (2017), showing similarities in patterns of bone strength between female rowers and early Holocene females; given the paucity of similar studies this would be a good one to cite here to support your suggestions

We thank reviewer 1 for the suggestion. We added the citation on page 6, lines 114-115.

page 7, lines 131-134: I think this should be reworded as a research question, or removed

We removed the sentence.

page 7, line 144: the term ‘nonathletic’ is throwing me a bit, is there a better word that could be used? Recreationally-active?

We replaced the term with the term nonathletes.

page 8: a visual diagram of sensor placement used in this study would be useful, even as Supporting Information

We created the diagram which is in supplementary figure 1.

page 10, line 206: maxEMG and iEMG need to be defined at first use, we do not yet know what these are until the Variables section below

We replaced the variables names with their definition.

Supplementary Videos: these videos play with a weird neon green colour for me! Is this perhaps just something wrong with the upload? It only goes to normal coloration towards the end of the videos. Also: it looks like these videos are of a male, given that your participants were all female, who is this male? Videos of your actual female participants doing this seem more appropriate.

We were not able to replicate the problem with video coloration, even when downloaded from the attached files. The male in the videos is one of the authors. Unfortunately, the videos were recorded after the experiments were finished.

page 12, line 253: In this paragraph, the description of ‘28.2 %MVC lower maxEMG’ is confusing, and rewording would help throughout this section—maybe something like ‘the posterior deltoid of athletes contracted with 28.2% less of its maximum voluntary contraction force than the posterior deltoid of controls’

We replaced the difference between means of athletes and nonathletes with mean values for athletes and nonathletes.

page 20: increased muscle activation has been documented in order to decrease the stress per active fiber, so this could be explored with regards to higher activation in the nonathlete posterior deltoid (and general trend overall)

We thank the reviewer for the suggestion, unfortunately, we were not able to find any literature on this topic.

page 20-21, lines 359-374: The arguments in this paragraph need more detail backing them up. Though CSA is a good proxy for force production, this is also affected by a huge range of factors like muscle length, muscle fiber type, concentration of muscle fibers per unit area, neuromuscular control etc. These factors affecting variation should be included in the discussion here, as it is too simplistic to conclude that lower activation of the upper limb muscles in athletes than non-athletes might be caused by the athletes simply having bigger muscles. Further, these trends of lower activation are for the most part non-significant, so that should be explicitly clear. I don’t think the authors can state that “this difference would not have an impact on muscle force stressing the upper limb bones” page 20, line 365) without providing some support for this statement from the literature or their own data. I also think that the concluding statement of this paragraph, “athletes would produce the same muscle force as nonathletes. Therefore, either of the samples may be used for the estimation of upper limb loading during grinding, if muscle hypertrophy is considered” is conjecture at the moment, as it is not backed up by citations or any evidence from the literature.

We added other factors into discussion on page 23, lines 446-448. We removed statements mentioned by the reviewer 1 and added explicit description of nonsignificant differences.

page 21, line 384: “upper limb muscle loading”- What does this mean? Is this a measured variable? Do you mean muscle activation?

• same applies to line 388 and 404- Can you clarify what ‘upper limb loading’ means more specifically? The muscles are less activated?

We replaced the ambiguous term with more explicit “approximation of muscle force on upper limb”.

page 22, line 415: what movement does the triceps brachii muscle do? Why would it be more involved in saddle quern grinding?

We added discussion of the role of triceps in elbow extension and stabilization (page 26, lines 518-521).

Reviewer #2:

In watching the cool videos of the grinding experiments I noticed the upper stone used with the saddle quern seemed unusually small. Typically these stones are about the width of the lower stone. The smaller stone would require a lot less force which could impact the utility of the experiments. Also while the text says grain was used in the experiments there is no further discussion of that. The grain on the grinding surfaces has a direct effect on the friction and thus the force required to move the stone. There is also no discussion of the manipulation of the grain. Grinders have to constantly stop and add and remove grain and would affect the experimental rhythm. Ideally the authors may have measured the work or forces involved in the movement of the machines, if so these data should be included. Otherwise they should at least provide the weight and dimensions of the upper stone and discuss the grain used in the experiment and its manipulation. Also I didn't see where they identified the material type of the stones. Actually grinding grain requires substantially more work that simply simulating the movement by moving the stone back and forth and would or course effect muscle activation. I am less familiar with rotary querns but some of these issues would apply, certainly the friction between the two grinding faces is an important part of the physics. Addressing these issues would foreclose concerns with these experiments.

We added description of the grain that was used in the experiment (page 10, lines 214-216). We also clarified that there was no manipulation of the grain during each grinding trial (page 11, lines 239-240 and 243). As suggested, we added dimensions and mass of the upper stone of the saddle quern (page 10, lines 208-210). We also added description of the materials that the querns are made of (page 10, lines 207-208 and 212-213).

Reviewer #3:

This is an interesting study. My only comment is with regard to the age and body size differences between the athletes and non-athletes. The athletes are younger, taller, and heavier. Would this not indicate they may not require as much muscle activation to produce the same force? However, I don't think it would have any impact on the results of your study.

Increased mass of rowers could have influence on muscle activation if they had more muscle mass. We are discussing this possible influence on pages 22-23 on lines 435-449. It is unlikely that the lower age of rowers had influence on muscle activation. Even if rowers were so young that they had less muscle mass than nonathletes, it would lead to rowers having higher muscle activation during grinding than nonathletes, which we did not observe.

---

## [Decision Letter · Decision Letter 1]

18 Aug 2021

Influence of upper limb training and analyzed muscles on estimate of physical activity during cereal grinding using saddle quern and rotary quern

PONE-D-20-36922R1

Dear Dr. Struška,

We’re pleased to inform you that your manuscript has been judged scientifically suitable for publication and will be formally accepted for publication once it meets all outstanding technical requirements.

Kind regards,

Michael D. Petraglia, Ph.D.

Academic Editor

PLOS ONE

Additional Editor Comments (optional):

Reviewers' comments:

Reviewer's Responses to Questions

**Comments to the Author**

1. If the authors have adequately addressed your comments raised in a previous round of review and you feel that this manuscript is now acceptable for publication, you may indicate that here to bypass the “Comments to the Author” section, enter your conflict of interest statement in the “Confidential to Editor” section, and submit your "Accept" recommendation.

Reviewer #3: All comments have been addressed

2. Is the manuscript technically sound, and do the data support the conclusions?

Reviewer #3: Yes

3. Has the statistical analysis been performed appropriately and rigorously? 

Reviewer #3: Yes

4. Have the authors made all data underlying the findings in their manuscript fully available?

Reviewer #3: Yes

5. Is the manuscript presented in an intelligible fashion and written in standard English?

Reviewer #3: Yes

6. Review Comments to the Author

Reviewer #3: line 90 seems to be a random leftover sentence. Line 389 - need space between nonathlete and could. No other comments

7. PLOS authors have the option to publish the peer review history of their article (what does this mean?). If published, this will include your full peer review and any attached files.

Reviewer #3: No

---

## [Editor Report · Acceptance letter]

23 Aug 2021

PONE-D-20-36922R1 

Influence of upper limb training and analyzed muscles on estimate of physical activity during cereal grinding using saddle quern and rotary quern 

Dear Dr. Struška:

I'm pleased to inform you that your manuscript has been deemed suitable for publication in PLOS ONE. Congratulations! Your manuscript is now with our production department. 

Kind regards, 

on behalf of

Professor Michael D. Petraglia 

Academic Editor

PLOS ONE